# Role of Artificial Intelligence Interpretation of Colposcopic Images in Cervical Cancer Screening

**DOI:** 10.3390/healthcare10030468

**Published:** 2022-03-03

**Authors:** Seongmin Kim, Hwajung Lee, Sanghoon Lee, Jae-Yun Song, Jae-Kwan Lee, Nak-Woo Lee

**Affiliations:** 1Gynecologic Cancer Center, CHA Ilsan Medical Center, CHA University College of Medicine, 1205 Jungang-ro, Ilsandong-gu, Goyang-si 10414, Korea; naiad515@gmail.com; 2Department of Obstetrics and Gynecology, Korea University College of Medicine, 73 Inchon-ro, Seongbuk-gu, Seoul 02841, Korea; ifigured35@gmail.com (H.L.); mdleesh@gmail.com (S.L.); jklee38@korea.ac.kr (J.-K.L.); nwlee@korea.ac.kr (N.-W.L.)

**Keywords:** artificial intelligence, cervical cancer screening, colposcopy, deep learning, machine learning

## Abstract

The accuracy of colposcopic diagnosis depends on the skill and proficiency of physicians. This study evaluated the feasibility of interpreting colposcopic images with the assistance of artificial intelligence (AI) for the diagnosis of high-grade cervical intraepithelial lesions. This study included female patients who underwent colposcopy-guided biopsy in 2020 at two institutions in the Republic of Korea. Two experienced colposcopists reviewed all images separately. The Cerviray AI^®^ system (AIDOT, Seoul, Korea) was used to interpret the cervical images. AI demonstrated improved sensitivity with comparable specificity and positive predictive value when compared with the colposcopic impressions of each clinician. The areas under the curve were greater with combined impressions (both AI and that of the two colposcopists) of high-grade lesions, when compared with the individual impressions of each colposcopist. This study highlights the feasibility of the application of an AI system in cervical cancer screening. AI interpretation can be utilized as an assisting tool in combination with human colposcopic evaluation of exocervix.

## 1. Introduction

Cervical intraepithelial neoplasia (CIN) is a premalignant lesion that is diagnosed and categorized as CIN1, CIN2, or CIN3 [1]. Genital human papillomavirus (HPV) infection is known as the critical step in the development of CIN [2]. If CIN is untreated, some patients may develop cervical cancer [3]. A diagnosis of CIN2-3 is a histological diagnosis obtained from biopsies of the suspect lesions, either with or without colposcopy, for which treatment is recommended. Screening for CIN can be achieved by cytological examination, human papillomavirus (HPV) screening, or colposcopy [4]. Among these, primary HPV testing is the most preferred method globally [5]. Regular screening for cervical cancer may lower the lifetime risk of the disease [6]. However, screening programs in low-income countries are difficult due to inaccessibility, lack of funding, lack of public policies, and high costs [7].

Colposcopy is used to identify cervical lesions using low-magnification microscopy with acetic acid and Lugol’s solution. It carries a sensitivity of 66–96% and specificity of 35–98% in diagnosing cervical lesions [8,9,10]. However, its accuracy varies according to the physician’s skill or proficiency [11].

The use of artificial intelligence (AI) in the medical field can improve the quality of care and cost-effectiveness [12]. Although machine learning can process a large amount of data in a relatively short time and has been successfully applied in many clinical situations, effective utilization of machine learning in actual clinical practice remains difficult [13]. Several studies have demonstrated the feasibility of clinical applications of AI in improving the diagnostic quality in CIN [14,15,16,17]. Previous studies evaluated the diagnostic value of AI for the interpretation of cervical images compared to that of cytology or histology.

The purpose of this study was to evaluate the feasibility of an AI system as an assistant tool in diagnosing high-grade CIN lesions compared to human interpretation of cervical images.

## 2. Materials and Methods

### 2.1. Study Patients and Terminology

This study included female patients who underwent colposcopy-guided biopsy because of abnormal cervical cytology or a positive HPV status during 2020 at two institutions located in Goyang and Seoul, Korea. Patients younger than 20 years or older than 50 years were excluded from the study. Additionally, unsatisfactory colposcopic images because of poor focus or invisible transformational zone were excluded from the study. Patient data along with cytologic and histopathological results following the biopsy were required for inclusion in the study. The cytological results in the data include either conventional Pap smear or liquid-based cytology. The histological results were obtained from the pathologic report from the biopsy, which was diagnosed by a professional pathologist in both institutions. Colposcopic images only included the cervical images with acetic acid applied on the cervix; images with Lugol’s solution applied on the cervix were not included. This study was approved by the institutional review board (2019AN0019). Bethesda classification system and CIN classification system were used for cytologic and histologic evaluation, respectively. The International Federation for Cervical Pathology and Colposcopy terminology was used for determining colposcopic impression.

### 2.2. Preparation of Machine Learning System

To interpret the cervical imaging, the Cerviray AI^®^ machine learning system (AIDOT, Seoul, Korea) was used, constructed with over 10,000 colposcopic images that were introduced to the learning algorithm along with histopathological diagnoses and clinical impressions of three gynecologic experts in colposcopy. A multi-category deep learning method was used by integrating (1) a knowledge-based clinical decision support system (CDSS) using the clinical colposcopic findings and histopathological results, and (2) non-knowledge-based CDSS via machine learning. The results interpreted by AI were classified as normal, CIN1, CIN2-3, or cancer. Figure 1 illustrates the interpretation of images using Cerviray AI^®^ deep learning system, which is composed of three main modules as follows:(1)Satisfactory filtering module was introduced to differentiate whether the taken colposcopic image is adequately satisfied for screening. This module is implemented by a convolutional neural network (CNN)-based classification model, which was trained to yield binary results that consist of satisfactory and unsatisfactory.(2)Preprocessing and normalization module was applied to prepare and adjust the image before AI interpretation. Colposcopic images are usually captured in uncontrolled environments, which result in various quality of the taken images such as poor contrast, brightness, etc. To compensate and improve the quality of the images, an auto-adjustment algorithm was implemented to preprocess and normalize them by applying various thresholding and filtering methods.(3)Feature extraction and cervical cancer diagnosis module have an important role in exploring the regions of the colposcopic images which correspond to suspicious precancerous cervical lesions. This module is implemented by CNN-based multi-class detection model named AIDOTNet v1.2, which was trained with multi-category images that consists the location of low and high-grade lesions. AIDOTNet v1.2 utilizes a pre-trained model to extract the suspicious region from a given image for predicting the lesion location in the image. In other words, the model leverages the feature extraction from the pre-trained model to locate the suspicious lesion box in the image and finally classifies the detected box as CIN1, CIN2-3, or cancer lesion. However, if no suspicious lesion box is detected from the colposcopic image, the model will yield normal as the AI interpretation result.

### 2.3. Clinical Interpretation of Colposcopic Finding

Two gynecologic oncologists separately examined all the images. Colposcopic impressions were divided into “non-specific”, “minor”, “major”, or “suspicious for invasion”. Multiple images of each patient were evaluated for an accurate diagnosis.

### 2.4. Statistical Analysis

Statistical analysis was performed using SPSS version 22.0 (IBM Inc., Armonk, NY, USA). The Kolmogorov–Smirnov test was used to verify the assumptions of the standard normal distributions. The Student’s t-test and Mann–Whitney U test were used to analyze the parametric and non-parametric variables, respectively. Differences between proportions were compared using Fisher’s exact test or χ^2^ test. Statistical significance was set at *p* < 0.05. Diagnostic accuracy was compared in terms of the sensitivity, specificity, and positive predictive value (PPV) between the cytological findings, colposcopic impressions, AI interpretations, and histopathological results. Pearson’s correlation coefficient was used to compare the correlations between the diagnostic tools. The accuracy of the diagnoses was evaluated in the validation set using receiver-operating characteristic (ROC) curves, which were created by plotting sensitivity against the false positive rate and its summary statistic, the area under the curve (AUC).

## 3. Results

### 3.1. Patient and Disease Characteristics

Overall, 234 patients were included in this study. The characteristics of the study population and diseases are presented in Table 1. Atypical squamous cells of unknown significance (ASC-US) were the commonest cytological result. The most frequent histological diagnosis was CIN2-3 followed by CIN1, benign findings including chronic cervicitis or koilocytotosis, and invasive cervical cancer. Almost half of the patients did not require any treatment; however, most of the patients with high-grade lesions were treated with conization or loop electrosurgical excision procedure (LEEP).

### 3.2. Evaluation of Diagnostic Accuracy

The distributions of impressions with each diagnostic tool according to the cytologic results are summarized in Table 2. ASC-US cytology resulted in various histological diagnoses, including benign lesion, CIN1, CIN2-3; otherwise, low-grade squamous intraepithelial lesion (LSIL) and high-grade squamous intraepithelial lesion (HSIL) cytology mostly resulted in corresponding histology.

The sensitivity, specificity, and PPV of each diagnostic tool are summarized in Table 3. AI demonstrated improved sensitivity with similar specificity and PPV compared with the colposcopic impression of each clinician. The sensitivity improved when the impressions of the two modalities were combined with at least one tool reporting suspicious high-grade lesions. The specificity of cytology was the highest among the tools compared.

Figure 2 illustrates the ROC curves for each diagnostic performance. AI demonstrated a higher AUC than Doctor 2 and a lower AUC than Doctor 1. However, if impressions of high-grade lesions were combined from the AI system and each Doctor, the AUCs improved compared with those of each clinician’s impressions.

### 3.3. Correlation between Diagnostic Performances

Figure 3 presents the correlation coefficients for each diagnostic tool. Doctors 1 and 2 demonstrated the highest correlation coefficients. However, cytology demonstrated a generally low correlation with other diagnostic tools.

## 4. Discussion

Colposcopy and directed biopsy are currently the major methods employed for diagnosing precancerous cervical lesions. However, several studies have demonstrated that even clinicians who are proficient in colposcopy have difficulties in making the correct diagnosis [18]. Therefore, the standardized and less fluctuating diagnostic performance of AI could play a role in this area. The feasibility of using deep learning-based colposcopy as an assistive diagnostic tool in high-grade CIN was evaluated in this study. The sensitivity of colposcopists in diagnosing CIN reportedly varies widely [19]. An inexperienced individual may miss high-grade lesions. Using the AI system, a non-professional gynecologist or general physician can make effective decisions regarding interventions (whether to perform a punch biopsy or transfer the patient to a specialized center).

The Cerviray^®^ (AIDOT) system achieved a better sensitivity and comparable PPV in predicting high-grade lesions compared with the gold standard evaluation method for biopsy based on colposcopy. This level of diagnostic accuracy was comparable to that reported in a large cohort study [20]. As demonstrated previously, AI interpretation includes better AUC in differentiating high-risk and low-risk lesions than the human interpretations of colposcopic images by both clinicians. Consequently, these results suggest that deep learning-based AI interpretations may be utilized in clinical use. This is also supported by a recent study that evaluated deep learning models to automatically classify colposcopic images [21]. The authors concluded that an improved AUC was observed using a machine learning-based system in discriminating high-grade lesions from low-grade lesions; therefore, AI systems may be suited for automated evaluations of colposcopic images. In another observational study, automated visual evaluation of cervical images demonstrated greater AUC than the original interpretation of cervical images by human or conventional cytology [15].

The results of this study show that even skilled colposcopists showed markedly increased sensitivity with the assistance of AI. In this study, if the colposcopists accepted the more aggressive impressions of AI despite disagreements with it, the AUC increased from 0.755 to 0.799 and 0.713 to 0.769 for Doctors 1 and 2, respectively. The sensitivity was also higher after acceptance of aggressive AI impression, in contrast to relatively low specificity and PPV after acceptance. Usually, high sensitivity is related to high negative predictive value (NPV) rather than PPV. The screening tools usually favors the diagnostic method, which shows high sensitivity and NPV. The Cerviray AI^®^ system was developed with the intention of utilizing the AI system in combination with human interpretation for screening high-grade cervical abnormality. Therefore, these subtle impairments of PPV might be acceptable.

Interestingly, as presented in Figure 3, the correlations between the two colposcopists were higher than any other correlations between the other modalities. AI interpretation and human colposcopic impressions demonstrated statistically significant correlations but a lower Pearson’s R than that between the two doctors. This observation implies that the AI system interprets colposcopic images using logic that is different from that is used in human colposcopic evaluations. The conventional colposcopic evaluation includes a triad of mosaic, punctuation, and aceto-white epithelium, which could be present as a mixture in a majority of cases with severe lesions [22]. In contrast, the Cerviray AI^®^ (AIDOT) system trains images under a subdivided network of serial processes (Figure 4). This process does not appear to follow the human colposcopy training but may include more delicate segmentation of abnormal lesions. Therefore, AI interpretations could be different from those of humans, but the logic for such interpretations remains unknown.

On the other hand, considering that the diagnostic value of AI interpretation was comparable to the impressions of colposcopic experts, AI interpretation might have a role as a diagnostic tool in evaluating high-grade cervical lesions in the distant future, especially in countries where certified or proficient colposcopists are insufficient. Generally, colposcopic evaluation includes a learning curve in achieving proficiency [23]. However, the AI system does not require this learning period, and this approach could improve the accessibility to cervical disease screening programs in developing countries or undeveloped countries. In the case of cytology and HPV testing, high lab equipment costs are incurred, and to operate the lab, it needs to build a lab and requires manpower, including pathologists, so there would be lots of operating costs. Therefore, it is recommended to use “visual Inspection with Acetic-acid” in underdeveloped areas, in which it is difficult to have cervical cancer screening [24,25]. Cerviray AI^®^ does not need special maintenance or training cost to use. Even if there are no specialists for diagnosis, patients can get a diagnosis from doctors through a telemedicine system. Therefore, it is a very efficient and useful device, especially in underdeveloped or developing countries.

Only a few previous studies have reported the feasibility of machine learning applications in colposcopic classification for cervical lesions. The accuracy of the validation dataset has been reported to be approximately 50% in classifying CIN3, carcinoma in situ, and invasive cancer in 158 patients who underwent conization [26]. Although the study demonstrated the feasibility of the AI application, it did not provide satisfactory accuracy. In another investigation with 170 images, an accuracy of 72% was reported in classifying the colposcopic images [27]. However, the clinical significance of those results is limited because only 58 images were used for training the machine learning system. Recently, a large-scale study in 9406 women reported that better diagnostic accuracy was observed with an automated visual evaluation using a deep learning-based AI system compared with the human interpretations or conventional cytology [15]. Cho et al. also evaluated deep learning models in automatically classifying cervical neoplasms using colposcopic photographs [21]. AI demonstrated a superior AUC over human colposcopic impressions. These previous studies have limitations in that the colposcopic findings were retrospective data derived from multiple colposcopists with varying experiences. However, in this study, all images were reviewed separately by two experienced colposcopists for the purposes of this study. This approach provides important information about the validation of the accuracy of human colposcopic impressions. It also enables a direct comparison of AI interpretations with colposcopic findings.

However, this study has a few limitations. Firstly, patients with atypical glandular cells were excluded from the study population due to the possible association with endometrial disease [28]. Secondly, colposcopic images only provide visual information of the exocervix; therefore, patients with endocervical lesions are not considered good candidates for accurate AI interpretations. Inadequate colposcopic finding usually requires additional endocervical evaluations, including endocervical cytology or endocervical curettage. We should not overlook the limitation of colposcopy itself in terms of the possibility that the transformation zone could be multifocal and could be hardly assessed while lying in the isthmus of the uterus or in the fornix of the vagina. Thirdly, there was heterogeneity in the image quality or resolution between patients due to the retrospective nature of the study. Fourthly, the human colposcopic impressions in this study may not reflect the real-time colposcopic diagnoses. Two colposcopists in this study evaluated only the digitalized images retrospectively. Real-time colposcopic diagnosis is based on a combination of visualization of abnormal patterns and rate of acetowhite changes, subtle differences in the degree of acetowhite response, and even the degree of light reflection. Therefore, the sensitivity and specificity of two colposcopists in this study should not be considered as a conventional colposcopic evaluation. Prospective studies to compare real-time colposcopic impressions and concomitant AI interpretations are warranted to address this issue. Fifthly, the presented sensitivity of cytology in Table 3 is relatively low. However, this shows a sensitivity at cutoff cytological high-grade lesions, including ASC-H or HSIL, for detection of histological CIN2 or worse. This could be a reason why the sensitivity is low in this study. In a meta-analysis, the sensitivity of liquid-based cytology and conventional cytology for CIN2 or worse showed 57.1 and 55.2%, respectively [29]. Additionally, the study population is not balanced between groups. The study population of this study were mostly received colposcopic evaluation because of an abnormal cytologic result or positive HPV testing. The low percentage of individuals with normal cervix could alter the diagnostic value. Finally, the percentage of histological CIN2-3 in ASCUS and LSIL cytology results is relatively high. However, there also exist which shows similar findings with this study. It is reported that 17–36% of patients with ASCUS cytology were diagnosed to have CIN2-3 on biopsy, and 34–50% of patients with LSIL cytology had CIN2-3 on biopsy [30]. However, we could deny that the ratio of CIN2-3 from ASCUS and LSIL is relatively high in this study. This could be because of a high proportion of patients who are positive for high-risk HPV. This also shows the importance of the HPV test for cervical cancer screening. The study population had cytology for their cervical cancer screening. The updated recommendation of primary HPV testing for cervical cancer globally should be considered, and further study from individuals with regular HPV testing should be performed later.

## 5. Conclusions

In conclusion, our study highlights the feasibility of using machine learning-based AI systems in cervical cancer screening. AI interpretation of cervical images could be an assistive tool if it is used in combination with human colposcopic evaluation. Additionally, if additional supportive studies are followed, it might be utilized as an alternative tool in evaluating high-grade cervical lesions when proficient colposcopists are unavailable due to the lack of accessibility or high cost in low-income or developing countries. Much more data are warranted for using AI systems in the field of cervical cancer screening.

## Figures and Tables

**Figure 1 healthcare-10-00468-f001:**
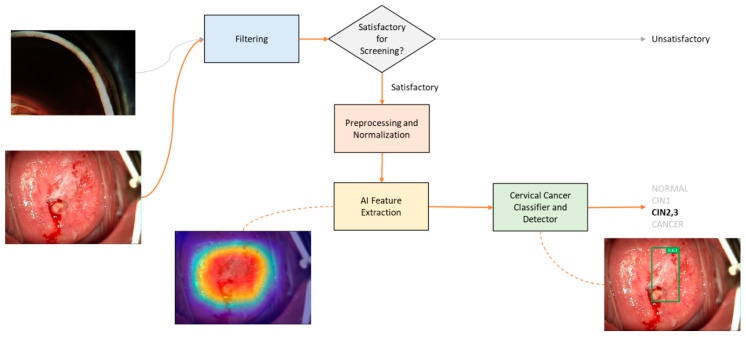
A diagram of Cerviray AI^®^ interpretation for colposcopic images. The system assesses the visibility of the images, and recognizes the squamocolumnar junction and transformation zone of the uterine cervix. If the image is satisfactory for evaluation, the image is processed and normalized for AI feature extraction. This is followed by the classification of images according to the AI impression.

**Figure 2 healthcare-10-00468-f002:**
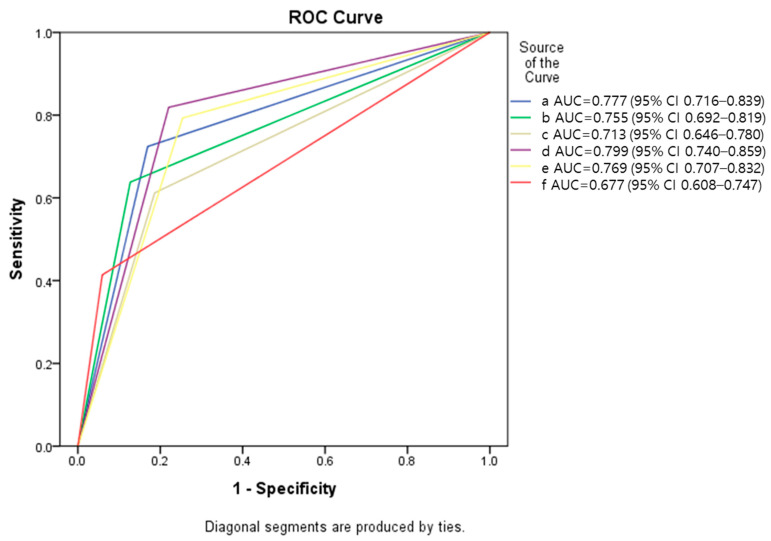
ROC curves of each diagnostic performance for detecting high-grade or worse lesion versus less severe impressions. (a) AI interpretation; (b) colposcopic impression of Dr 1; (c) colposcopic impression of Dr 2; (d) combined impression of AI and Dr1 colposcopy; (e) combined impression of AI and Dr2 colposcopy; (f) cytology.

**Figure 3 healthcare-10-00468-f003:**
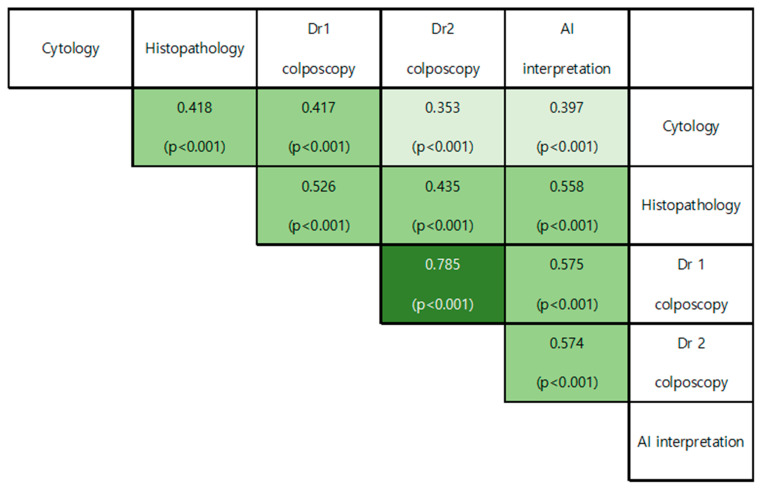
Correlations between each diagnostic tool. Values are expressed as Pearson’s R (*p*-value).

**Figure 4 healthcare-10-00468-f004:**
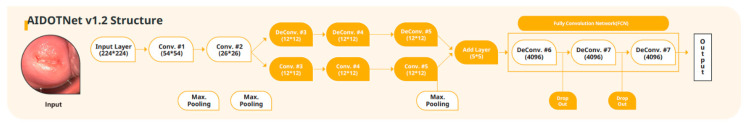
An algorithm of the deep learning process of Cerviray AI^®^. Briefly, it included the input of an image, multiple convolution and deconvolution networks of image processing while pooling and dropping out of data, and output of the result.

**Table 1 healthcare-10-00468-t001:** Clinical Characteristics of Study Population.

Characteristics	Value
Age, years	36.9 ± 8.9
Cytological results	
Normal	5 (2.1)
ASC-US	107 (45.7)
LSIL	67 (28.6)
ASC-H/HSIL	52 (22.2)
SCC	3 (1.3)
HPV status	
Positive for high-risk	153 (65.4)
Positive for low-risk only or negative	16 (6.8)
Not done	65 (27.8)
Histopathology	
Benign	52 (22.2)
CIN1	66 (28.2)
CIN2-3	110 (47.0)
Invasive cancer	6 (2.6)
Treatment	
Observation and follow-up	111 (47.4)
LEEP/Conization	107 (45.7)
Extrafascial hysterectomy	5 (2.1)
Radical hysterectomy	4 (1.7)
Chemotherapy/Radiotherapy	2 (0.9)
Refusal of treatment	5 (2.1)

Values are expressed as mean ± standard deviation or number (%). ASC-H: atypical squamous cells, cannot exclude high-grade squamous intraepithelial lesions; ASC-US, atypical squamous cells of unknown significance; CIN, cervical intraepithelial neoplasia; HSIL, high-grade squamous intraepithelial lesion; HPV, human papilloma virus; LEEP, loop electrosurgical excision procedure; LSIL, low-grade squamous intraepithelial lesion.

**Table 2 healthcare-10-00468-t002:** Distribution of the colposcopic findings, AI interpretations, and histopathology according to the cytology results.

Cytology	Impression	Doctor 1	Doctor 2	AI	Histopathology
Normal	Non-specific/Benign	2	2	3	4
	Minor/CIN1	2	3	2	0
	Major/CIN2-3	1	0	0	1
ASC-US	Non-specific/Benign	28	35	43	37
	Minor/CIN1	50	32	30	34
	Major/CIN2-3	32	39	32	35
	Suspicious for invasion/Cancer	0	1	2	1
LSIL	Non-specific/Benign	15	14	20	7
	Minor/CIN1	37	32	24	29
	Major/CIN2-3	15	21	22	31
	Suspicious for invasion/Cancer	0	0	1	0
ASC-H/ HSIL	Non-specific/Benign	4	4	7	4
	Minor/CIN1	6	9	5	3
	Major/CIN2-3	41	38	37	43
	Suspicious for invasion/Cancer	1	1	3	2
SCC	Suspicious for invasion/Cancer	3	3	3	3

Values are expressed as a number. AI, artificial intelligence; ASC-H, atypical squamous cells, cannot exclude high-grade squamous intraepithelial lesions; ASC-US, atypical squamous cells of unknown significance; CIN, cervical intraepithelial neoplasia; HSIL, high-grade squamous intraepithelial lesion; LSIL, low-grade squamous intraepithelial lesion; SCC, squamous cell carcinoma.

**Table 3 healthcare-10-00468-t003:** Evaluation of the diagnostic quality of various tools in detecting high-grade or worse lesions versus less severe impressions.

Method	Sensitivity	Specificity	PPV
Cytology	41.38	94.07	87.27
Doctor 1	71.55	87.29	84.69
Doctor 2	69.83	81.36	78.64
AI interpretation	74.14	83.05	81.13
Doctor 1 + AI	84.48	77.97	79.03
Doctor 2 + AI	83.62	74.58	76.38

AI, artificial intelligence; PPV, positive predictive value; Doctor 1 + AI, if Doctor 1 accepted the more aggressive impressions of AI despite disagreements; Doctor 2 + AI, if Doctor 2 accepted the more aggressive impressions of AI despite disagreements.

## Data Availability

The data presented in this study are available on request from the corresponding author.

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
