# Peer review of "Role of Artificial Intelligence Interpretation of Colposcopic Images in Cervical Cancer Screening"

_healthcare, 2022, doi:10.3390/healthcare10030468_

Round 1

Reviewer 1 Report

This manuscript portrays application of an exist AI system to cervical images for detection of high-grade cervical intraepithelial lesions. Few queries are:

  1. As study does not include development of a new AI system and neither is first in the field to use AI. It seems more of validation of an existing AI system than a pilot study.
  2. In methodology section, authors state “To construct this system, over 10,000 colposcopic im- 67 ages were introduced to the learning algorithm along with histopathological diagnoses 68 and clinical impressions of three gynecologic experts in colposcopy.” Source and processing of these 10000 images should be discussed in detail. If this system is part of “Cerviray AI® machine learning system (AIDOT, 66 Seoul, Republic of Korea) ”, then appropriate reference should be provided pointing to development and validation of that system.
  3. Statistical significance between group should mentioned for difference in sensitivity, specificity and PPV.
  4. In discussion section, para 3; authors state : “The results of this study shows that even skilled colposcopists could increase their 170 diagnostic accuracy with markedly increased sensitivity with the assistance of AI. In this 171 study, if the colposcopists accepted the more aggressive impressions of AI despite disa- 172 greements with it, the AUC increased from 0.755 to 0.799 and 0.713 to 0.769 for Doctors 1 173 and 2, respectively.” Here, Sensitivity should not be confused with diagnostic accuracy. 
  5. While sensitivity of the AI+Dr was more than clinician alone but PPV was less. Same should be discussed in results and discussion.
  6. Given large data and previously published studies (ref 14-17 in manuscript), what is the new message in the current study? 

Author Response

Responses to reviewer 1 [healthcare-1589779]

Title: Role of artificial intelligence interpretation of colposcopic images in cervical cancer screening

I really appreciate for your review of the manuscript. I have applied all the possible changes you recommended. It was an honor for me to learn your professional point of view regarding detail approach for writing an original article. I think this experience would improve my future works.

Thank you.

Best Regards,

Seongmin Kim

This manuscript portrays application of an exist AI system to cervical images for detection of high-grade cervical intraepithelial lesions. Few queries are:

1. As study does not include development of a new AI system and neither is first in the field to use AI. It seems more of validation of an existing AI system than a pilot study.

: Thank you for your comment. ‘A pilot study’ has been removed from the title of the study.

2. In methodology section, authors state “To construct this system, over 10,000 colposcopic im- 67 ages were introduced to the learning algorithm along with histopathological diagnoses 68 and clinical impressions of three gynecologic experts in colposcopy.” Source and processing of these 10000 images should be discussed in detail. If this system is part of “Cerviray AI® machine learning system (AIDOT, 66 Seoul, Republic of Korea) ”, then appropriate reference should be provided pointing to development and validation of that system.

: Figure 1 illustrates the interpretation of images using Cerviray AI®. deep learning system which is composed of three main modules as follows:

1)           Satisfactory filtering module was introduced to differentiate whether the taken colposcopic image is adequately satisfied for screening. This module is implemented by a Convolutional Neural Network (CNN)-based classification model which was trained to yield binary results that consists of satisfactory and unsatisfactory.

2)           Preprocessing and normalization module was applied to prepare and adjust the image before AI interpretation. Colposcopic images are usually captured in uncontrolled environments, which result in various quality of the taken images such as poor contrast, brightness, etc. To compensate and improve the quality of the images, an auto-adjustment algorithm was implemented to preprocess and normalize them by applying various thresholding and filtering methods.

3)           Feature extraction and cervical cancer diagnosis module has an important role in exploring the regions of the colposcopic images which correspond to suspicious pre-cancerous cervical lesion. This module is implemented by CNN-based multi-class detection model, named AIDOTNet v1.2, which was trained with multi-category images that consists the location of low and high-grade lesions. AIDOTNet v1.2 utilizes a pre-trained model to extract the suspicious region from a given image for predicting the lesion location in the image. In other words, the model leverages the feature extraction from the pre-trained model to locate the suspicious lesion box in the image, and finally classifies the detected box as CIN1, CIN2-3, or cancer lesion. However, if no suspicious lesion box is detected from the colposcopic image, the model will yield normal as the AI interpretation result.

3. Statistical significance between group should mentioned for difference in sensitivity, specificity and PPV.

: Thank you for your kind comment. However, I respectfully hope you to consider maintaining the current analysis. As you already know, not all studies comparing sensitivity or specificity show statistical significance for the purpose of comparison. Many studies which are already referenced in these studies neither included statistical significance in their analyses. I think the value of AUC and 95% CI is more valuable in comparison of screening tools. Therefore, I hope Figure 2 could substitute this requirement. Would you consider about this once more? Thank you again for your great review.

4. In discussion section, para 3; authors state : “The results of this study shows that even skilled colposcopists could increase their 170 diagnostic accuracy with markedly increased sensitivity with the assistance of AI. In this 171 study, if the colposcopists accepted the more aggressive impressions of AI despite disa- 172 greements with it, the AUC increased from 0.755 to 0.799 and 0.713 to 0.769 for Doctors 1 173 and 2, respectively.” Here, Sensitivity should not be confused with diagnostic accuracy. 

: Thank you. This sentence was modified to make the meaning of the terms more clear.

5. While sensitivity of the AI+Dr was more than clinician alone but PPV was less. Same should be discussed in results and discussion.

: I appreciate to the great point of your review. The sensitivity was also higher after acceptance of aggressive AI impression, in contrast to relatively low specificity and PPV after acceptance. Usaully, high sensitivity is related to high negative predictive value (NPV) rather than PPV. The screening tools usually favors diagnostic method which show high sensitivity and NPV. The Cerviray AI® system was developed with an intention that utilizing AI system in combination of human interpreta-tion for screening high-grade cervical abnormality. Therefore, these subtle impairment of PPV might be acceptable. This was stated in discussion section.

6. Given large data and previously published studies (ref 14-17 in manuscript), what is the new message in the current study? 

: Previous studies evaluated diagnostic value of AI for interpretation of cervical images compared to that of cytology or histology.

The purpose of this study was to evaluate the feasibility of an AI system as an assistant tool in diagnosing high-grade CIN lesions compared to human interpretation of cervical images.

Reviewer 2 Report

The work presented is well written and has interesting results in its field. The following are some recommendations:

1.The introduction should be improved. It is important for the reader to have a better context of the system they are using or bibliography to support this. Also, What is the difference between the studies in page 2 line 46 and your tool?

2. It is important that the materials and methods be given greater context. For example, where are the clinics located? 

3. AUC and PPV methods are adequate for the prediction they are making. It should be considered that many of the tests are not balanced and this affects the result. For example, in Table 2 we have a histopathology of 4, 0 and 1. 

4. Table 3 mentions Dr 1 + AI and Dr 2 + AI, how are they being combined? This question is so that I can better understand what was done.

5. The statistical analysis was well explained and has a good justification. My question is on the ML process. The content of the machine learning test methodology needs to be improved. How is the model trained? Sensitivity, specificity and PPV were drawn with 10 fold evaluation or in what way? How was the data training process?

These points need to be addressed for a better understanding of the process.

Author Response

Responses to reviewer 2 [healthcare-1589779]

Title: Role of artificial intelligence interpretation of colposcopic images in cervical cancer screening

I really appreciate for your review of the manuscript. I have applied all the possible changes you recommended. It was an honor for me to learn your professional point of view regarding detail approach for writing an original article. I think this experience would improve my future works.

Thank you.

Best Regards,

Seongmin Kim

The work presented is well written and has interesting results in its field. The following are some recommendations:

1.The introduction should be improved. It is important for the reader to have a better context of the system they are using or bibliography to support this. Also, What is the difference between the studies in page 2 line 46 and your tool?

:Thank you for your opinion. Previous studies evaluated diagnostic value of AI for interpretation of cervical images compared to that of cytology or histology.

The purpose of this study was to evaluate the feasibility of an AI system as an assistant tool in diagnosing high-grade CIN lesions compared to human interpretation of cervical images.

  1. It is important that the materials and methods be given greater context. For example, where are the clinics located? 

: Thank you for this comment. I have revised the materials and methods section to give more detail information.

  1. AUC and PPV methods are adequate for the prediction they are making. It should be considered that many of the tests are not balanced and this affects the result. For example, in Table 2 we have a histopathology of 4, 0 and 1. 

: Thank you. I totally agree with your opinion. The study population is not balanced between groups. The study population of this study were mostly received colposcopic evaluation because of an abnormal cytologic result or positive HPV testing. The low percentage of individuals with normal cervix could alter the diagnostic value. This was included in revised discussion section about limitation of the study.

  1. Table 3 mentions Dr 1 + AI and Dr 2 + AI, how are they being combined? This question is so that I can better understand what was done.

: I also thank you for this remark. The explanations were added in Table 3 legend. Doctor 1 + AI, if Doctor 1 accepted the more aggressive impressions of AI despite disagreements; Doctor 2 + AI, if Doctor 2 accepted the more aggressive impressions of AI despite disagreements.

  1. The statistical analysis was well explained and has a good justification. My question is on the ML process. The content of the machine learning test methodology needs to be improved. How is the model trained? Sensitivity, specificity and PPV were drawn with 10 fold evaluation or in what way? How was the data training process?

: Thank you for your comments. Figure 1 illustrates the interpretation of images using Cerviray AI®. deep learning system which is composed of three main modules as follows:

1)           Satisfactory filtering module was introduced to differentiate whether the taken colposcopic image is adequately satisfied for screening. This module is implemented by a Convolutional Neural Network (CNN)-based classification model which was trained to yield binary results that consists of satisfactory and unsatisfactory.

2)           Preprocessing and normalization module was applied to prepare and adjust the image before AI interpretation. Colposcopic images are usually captured in uncontrolled environments, which result in various quality of the taken images such as poor contrast, brightness, etc. To compensate and improve the quality of the images, an auto-adjustment algorithm was implemented to preprocess and normalize them by applying various thresholding and filtering methods.

3)           Feature extraction and cervical cancer diagnosis module has an important role in exploring the regions of the colposcopic images which correspond to suspicious pre-cancerous cervical lesion. This module is implemented by CNN-based multi-class detec-tion model, named AIDOTNet v1.2, which was trained with multi-category images that consists the location of low and high-grade lesions. AIDOTNet v1.2 utilizes a pre-trained model to extract the suspicious region from a given image for predicting the lesion location in the image. In other words, the model leverages the feature extraction from the pre-trained model to locate the suspicious lesion box in the image, and finally classifies the detected box as CIN1, CIN2-3, or cancer lesion. However, if no suspicious lesion box is detected from the colposcopic image, the model will yield normal as the AI interpretation result.

Reviewer 3 Report

The authors studied the role of artificial intelligence (AI) in its use for interpretation of colposcopic images in cervical cancer screening. This retrospective studied compared the response of Cerviray AI machine learning system (AIDOT, Seoul, Republic of Korea) against two experienced colposcopists. They concluded that the AI system achieved a better sensitivity and comparable PPV in predicted high grade lesions as compared to colposcopists. 

They discussed the limitations of their study in the discussion. The two colposcopists were only able to see digitised images retrospectively. They were not able to see any real-time images, or visualise the degree of acetowhite response. This essentially negates many of the advantages of having a human interpret the data as it takes away everything other than the digitised images. As such, it does not seem to be a fair comparison, and one wonders what the sensitivity and PPV comparison would look like if the colposcopists were allowed to review patients in real time. 

The argument that this may help in areas where there is low accessibility or high cost in low-income or developing countries runs into the issue of the cost of the AI system itself, the cost of maintenance and the training associated with using such a system and perhaps this should be mentioned. 

Author Response

Responses to reviewer 3 [healthcare-1589779]

Title: Role of artificial intelligence interpretation of colposcopic images in cervical cancer screening

I really appreciate for your review of the manuscript. I have applied all the possible changes you recommended. It was an honor for me to learn your professional point of view regarding detail approach for writing an original article. I think this experience would improve my future works.

Thank you.

Best Regards,

Seongmin Kim

The authors studied the role of artificial intelligence (AI) in its use for interpretation of colposcopic images in cervical cancer screening. This retrospective studied compared the response of Cerviray AI machine learning system (AIDOT, Seoul, Republic of Korea) against two experienced colposcopists. They concluded that the AI system achieved a better sensitivity and comparable PPV in predicted high grade lesions as compared to colposcopists. 

They discussed the limitations of their study in the discussion. The two colposcopists were only able to see digitised images retrospectively. They were not able to see any real-time images, or visualise the degree of acetowhite response. This essentially negates many of the advantages of having a human interpret the data as it takes away everything other than the digitised images. As such, it does not seem to be a fair comparison, and one wonders what the sensitivity and PPV comparison would look like if the colposcopists were allowed to review patients in real time. 

:Thank you for your comment. I have added that VIA images were used for this study.

The argument that this may help in areas where there is low accessibility or high cost in low-income or developing countries runs into the issue of the cost of the AI system itself, the cost of maintenance and the training associated with using such a system and perhaps this should be mentioned.

: Thank you for this comment. This could be an important point that AI system have benefits in terms of economic aspects.  In case of cytology and HPV testing, high lab equipment costs are incurred, and to operate the lab, it needs to build a lab and require manpower including pathologists so there would be lots of operating costs. Therefore, it is recommended to use VIA(Visual Inspection with Acetic-acid) in underdeveloped areas, which are difficult to have cervical cancer screening. Cerviray AI® does not need special maintenance or training cost to use. Only internet fee is required. Even if there are no specialists for diagnosis, patients can get diagnosis from doctors through telemedicine system. Therefore, it is a very efficient and useful device especially underdeveloped or developing countries.

This manuscript is a resubmission of an earlier submission. The following is a list of the peer review reports and author responses from that submission.

Round 1

Reviewer 1 Report

A special attention should be paid to the AI used for automatic interpretation of colposcopic images in the cervical cancer screening. Thus, a potential importance of the reviewed study arises.

At present stage of knowledge, there are two the most promising fields in an application of the AI in colposcopy. First, it could be used as a tool supporting an education process in colposcopy, the second one, potentially, it could be adopted in a quality and assessment control of the colposcopic procedure.

Please find below issues that should be clarified in each section of the manuscript:

Introduction

1) Please update your references in the study in a natural history of HPV infection.

2) Line 29

CIN2 and CIN3 are not “considered” high-lesions – they are high-grade lesions. Please, correct it.

3) “They are usually diagnosed based on the findings of biopsy and cytological examination of suspected lesions, human papillomavirus (HPV) screening, or colposcopy [3].

This sentence is unclear and could be misinterpreted. Please, rephrase it and split in two separate sentences about histological diagnosis of high-grade lesions based on findings in colposcopy with biopsy, and about screening of high-grade lesions. These are two different issues.

4) Furthermore, please specify the modalities used worldwide in cervical cancer screening globally, and include that HPV-based screening is globally recommended approach for all settings independent on the income (ASCO), as well. Please also include this issue in the discussion section and correlate with your approach that is a cytology-based.

5) “It carries a sensitivity of 70%–98% and specificity of 45%–90% in diagnosing cervical lesions [6-8].”

Please update the references No 6-7 about colposcopy diagnostic value – references from 90’ are not very accurate.

Methods

1) 234 patients were included from how large initial group, how about the inclusion criteria? % of referrals to colposcopy?

 2) AI in colposcopy is a new method that should be described in detail in a manuscript, according with good standards of publishing. In that matter:

  • please indicate a colposcopy protocol used in the study
  • please describe a preparation of the cervix for an image acquisition
  • it would be very valuable to present a comparison between real areas where biopsies were taken from vs. suspicious areas indicated by AI

3) Please specify, in accordance with which guidelines/terminology histologic diagnoses were reported.

4) What cytologic laboratory preparation was used in the study? By whom cytology was evaluated? There are no cytologic-virologic correlations mentioned, what would allow to a quality control insight for a gynecologic cytopathology, as well (due to very low sensitivity of cytology).

5) Indication of HRHPV14 status in the analyzed group would allow for an initial evaluation of AI interpretation in HPV-based model, that became a global standard of cervical cancer screening.

6) Figure 1 – what does it mean “satisfactory for screening” in a diagram of AI interpretation for colposcopic images? Please indicate what terminology was used in the study in the Methods and eventually correct it in Figure 1.

7) Lines 53 and 72-73

Lack of correspondence with IFCPC 2011 nomenclature for colposcopic lesions. Please specify what colposcopic terminology was used in the study.

Discussion

1) Despite of a method of colposcopic assessment used (by an expert or with AI application), it is crucial to understand the colposcopy limitations as a diagnostic procedure. It is not only for AGC cases but for all lesions developing in endocervical canal or in deep cervical crypts. It should be noted that the transformation zone could be multifocal and could be hardly assessed while lying in the isthmus of the uterus or in the fornix of the vagina – these issues are the starting points for a modern standardized colposcopy analysis. Please include these limitations of colposcopy and AI in the Discussion.

2) Presented sensitivity of cytology is very low – please clarify the reasons for such a low level of the sensitivity in the Discussion.

3) The percentage of histologic HSIL in ASC-US and LSIL (32.7% and 55.2%, respectively) cytology results was quite inconsistent with other international reports. Please include that issue in the Discussion.

4) “Two gynecologic oncologists separately examined all the images. Colposcopic im- 71 pressions were divided into normal, low-grade squamous intraepithelial lesion (LSIL), high-grade squamous intraepithelial lesion (HSIL), or cancer. Multiple images of each patient were evaluated for an accurate diagnosis.”

“Third, the human colposcopic impressions in this study may not reflect the real-time colposcopic diagnoses. Two colposcopists in this study evaluated only the digitalized images.”

Based on these two abovementioned paragraphs: please clarify, who has been performed colposcopy with biopsy in your study?

Who has been applied the AI?

5) “better sensitivity and comparable PPV in predicting high-grade lesions compared with the gold standard grading based on colposcopic impressions”

Histopathology is a gold standard, not a colposcopic impression.

6) “An inexperienced individual may miss high-grade lesions. Using the AI system, a non-professional gynecologist or general physician can make effective decisions regarding interventions (whether to perform a punch biopsy or transfer the patient to a specialized center).”

Authors are comparing and analyzing in the study a diagnostic value of colposcopy performed by experts in the field (gynecologic oncologists). While, in discussion it is described that AI can be used by unexperienced gynecologists and general physicians, as well. Please give a proper discussion on that matter that is in correlation with a study design and presented results as the analysis on high level of colposcopic expertise was compared with the AI application.

7) “However, the original intention of the Cerviray AI® system (AIDOT) was to use it for cervical cancer screening in low-income or developing countries where a large number of patients cannot access medical services due to the lack of accessibility or high costs. Therefore, the omission of patients with atypical glandular cells appears to be reasonable.”

It is not very good reason for the omission of AGC, as being country with a very high HDI. Please, explain only in the discussion the limitations of AI in detecting endocervical disease, not only with a glandular origin. Intraepithelial lesions can arise in endocervical canal as well. Is there any other limitation of AI in colposcopy?

Results

1) “Chronic cervicitis/koilocytic atypia

The histologic diagnosis of LSIL should be established if there was the koilocytic atypia in a morphologic view. Chronic cervicitis and koilocytic atypia have different morphologic features – these two are different histologic diagnoses. There is no chance to do a proper cytologic-histologic correlation using that kind of histologic diagnoses. Please also mention that issue in the Discussion as the limitation of your study.

Please specify also in the Methods, what histology terminology for cervical biopsies interpretation was used in the study. Please also make your introduction correspondent with your results in that matter (LSIL vs CIN terminology).

Material and methods vs results

1) “This study included female patients who underwent colposcopy-guided biopsy because of abnormal cervical cytology or a positive HPV status during 2020 at two institutions.”

Authors stated that the study included women with abnormal cervical cytology results or HPV-positive. While, further in the study there is any mention about patients with positive HPV status including in Patient and Characteristics subsection and table 1. If you exclude HPV-positive patients from the study, please mention this in the manuscript. That issue is not clarified enough in the study.

Conclusions

In the light of abovementioned issues in the review, presented conclusions are not cautious enough and are going too far in AI interpretation as a stand-alone tool in cervical cancer screening. First and foremost, conclusions are not based on presented results. Please, follow the purpose of your study in conclusions.

In summary, caution is needed for AI and precancer and cancer diagnosis. Much more data is needed before we can start using such systems in diagnostics.

Author Response

I really appreciate for your review of the manuscript. I have applied all the changes you recommended. It was an honor for me to learn your professional point of view regarding detail approach for writing review article. I think this experience would improve my future works.

Thank you.

Best Regards,

Seongmin Kim

A special attention should be paid to the AI used for automatic interpretation of colposcopic images in the cervical cancer screening. Thus, a potential importance of the reviewed study arises.

At present stage of knowledge, there are two the most promising fields in an application of the AI in colposcopy. First, it could be used as a tool supporting an education process in colposcopy, the second one, potentially, it could be adopted in a quality and assessment control of the colposcopic procedure.

Please find below issues that should be clarified in each section of the manuscript:

Introduction

  • Please update your references in the study in a natural history of HPV infection.

> I have updated the introduction section regarding your comment.

  • Line 29

CIN2 and CIN3 are not “considered” high-lesions – they are high-grade lesions. Please, correct it.

> Thank you for the comment. The sentence was removed and substituted with different expressions.

  • “They are usually diagnosed based on the findings of biopsy and cytological examination of suspected lesions, human papillomavirus (HPV) screening, or colposcopy [3]. “

This sentence is unclear and could be misinterpreted. Please, rephrase it and split in two separate sentences about histological diagnosis of high-grade lesions based on findings in colposcopy with biopsy, and about screening of high-grade lesions. These are two different issues.

> Thank you. The sentence was separated and modified regarding your advice.

  • Furthermore, please specify the modalities used worldwide in cervical cancer screening globally, and include that HPV-based screening is globally recommended approach for all settings independent on the income (ASCO), as well. Please also include this issue in the discussion section and correlate with your approach that is a cytology-based.

> Thank you for this opinion. I have specified about primary HPV testing in introduction section. And additional limitation of this study regarding this issue was included in discussion section.

5) “It carries a sensitivity of 70%–98% and specificity of 45%–90% in diagnosing cervical lesions [6-8].”

Please update the references No 6-7 about colposcopy diagnostic value – references from 90’ are not very accurate.

> The references were updated. Thank you.

Methods

  • 234 patients were included from how large initial group, how about the inclusion criteria? % of referrals to colposcopy?

> Initial number of study population is same with the number of patients included in the study. As mentioned in method section, the data of all patients who had colposcopy-guided biopsy of uterine cervix because of abnormal cytology or positive HPV testing was retrospectively collected and reviewed. Therefore, in this study population, 100% of patients had colposcopy.

  • AI in colposcopy is a new method that should be described in detail in a manuscript, according with good standards of publishing. In that matter:
  • please indicate a colposcopy protocol used in the study
  • please describe a preparation of the cervix for an image acquisition
  • it would be very valuable to present a comparison between real areas where biopsies were taken from vs. suspicious areas indicated by AI

> I want you to know that this study used retrospectively collected image data. If this study was a prospective setting, there could be a definite protocol for colposcopy and cervical preparation. And maybe, the location of biopsy could have been reported. However, only colposcopic images with acetic acid application from various gynecologist in two institutions were retrospectively collected and reviewed by two colposcopists in this study. This is the limitation of this study.

  • Please specify, in accordance with which guidelines/terminology histologic diagnoses were reported.

>Generally, the Bethesda system terminology was used in this study. However, the original CIN terminology was also used for histologic diagnosis, which could not be changed by this retrospective data review.

  • What cytologic laboratory preparation was used in the study? By whom cytology was evaluated? There are no cytologic-virologic correlations mentioned, what would allow to a quality control insight for a gynecologic cytopathology, as well (due to very low sensitivity of cytology).

>The cytology was performed at local clinic or my hospital during medical check-up. Therefore, the cytologic preparation included both conventional pap smear and liquid-based cytology. We cannot unify the screening tool of study population because the patients were refereed from various clinics. The evaluation of cytology is also performed by various pathologists. The histologic confirmation of biopsy was performed by various pathologists in two institutions. But this confirmation is not for this study. The pathologic reports were retrospectively reviewed for the study.

  • Indication of HRHPV status in the analyzed group would allow for an initial evaluation of AI interpretation in HPV-based model, that became a global standard of cervical cancer screening.

> I will consider this in further study.

  • Figure 1– what does it mean “satisfactory for screening” in a diagram of AI interpretation for colposcopic images? Please indicate what terminology was used in the study in the Methods and eventually correct it in Figure 1.

> The system assesses the visibility of images, recognizes squamocolumnar junction and transformation zone of uterine cervix. If the image is satisfactory for evaluation, the image is processed and normalized for AI feature extraction. The correction was made in Figure 1 legend.

7) Lines 53 and 72-73

Lack of correspondence with IFCPC 2011 nomenclature for colposcopic lesions. Please specify what colposcopic terminology was used in the study.

>Colposcopic terminology was not used for this study. “Colposcopic impression” does not refer to colposcopic lesions. It means an impression of colposcopist who retrospectively reviewed the images, and the classification of normal/LSIL/HSIL/cancer was used for the impression. This study is not a prospective study, so there was no unified protocol for conducting colposcopy.

Discussion

  • Despite of a method of colposcopic assessment used (by an expert or with AI application), it is crucial to understand the colposcopy limitations as a diagnostic procedure. It is not only for AGC cases but for all lesions developing in endocervical canal or in deep cervical crypts. It should be noted that the transformation zone could be multifocal and could be hardly assessed while lying in the isthmus of the uterus or in the fornix of the vagina – these issues are the starting points for a modern standardized colposcopy analysis. Please include these limitations of colposcopy and AI in the Discussion.

> Thank you for your opinion. Discussion section was modified regarding your comment.

  • Presented sensitivity of cytology is very low – please clarify the reasons for such a low level of the sensitivity in the Discussion.

> Presented sensitivity of Pap smear in Table 3 is sensitivity for histologic CIN2 or worse from HSIL or worse finding in cytology. It is not sensitivity for any abnormal histology from any abnormal cytology. This could be a reason why the sensitivity is low in this study.

3) The percentage of histologic HSIL in ASC-US and LSIL (32.7% and 55.2%, respectively) cytology results was quite inconsistent with other international reports. Please include that issue in the Discussion.

>There also exist which shows similar findings with this study. Ovestad et al. reported that 17-36% of patients with ASCUS cytology were diagnosed to have CIN2-3 in biopsy, and 34-50% of patients with LSIL cytology had CIN 2-3 in biopsy [1]. However, it is true that the ratio of CIN2/3 from ASCUS and LSIL is relatively high. This could be a high proportion of patients who have positive high-risk HPV. This also shows the importance of HPV test for cervical cancer screening.

[1]4) “Two gynecologic oncologists separately examined all the images. Colposcopic impressions were divided into normal, low-grade squamous intraepithelial lesion (LSIL), high-grade squamous intraepithelial lesion (HSIL), or cancer. Multiple images of each patient were evaluated for an accurate diagnosis.”

“Third, the human colposcopic impressions in this study may not reflect the real-time colposcopic diagnoses. Two colposcopists in this study evaluated only the digitalized images.”

Based on these two abovementioned paragraphs: please clarify, who has been performed colposcopy with biopsy in your study?

Who has been applied the AI?

>As previously replied, I want you to know that this study used retrospectively collected image data. The data of all patients who had colposcopy-guided biopsy of uterine cervix because of abnormal cytology or positive HPV testing was retrospectively collected and reviewed. Only patients with full data of cytology, histology, colposcopic image were included in the study, and all images were reviewed again by two colposcopists for this study, and applied AI interpretation.

5) “better sensitivity and comparable PPV in predicting high-grade lesions compared with the gold standard grading based on colposcopic impressions”

Histopathology is a gold standard, not a colposcopic impression.

> I understand that histologic confirmation is the only standard. What I meant was that the standard method for biopsy is colposcopy.

6) “An inexperienced individual may miss high-grade lesions. Using the AI system, a non-professional gynecologist or general physician can make effective decisions regarding interventions (whether to perform a punch biopsy or transfer the patient to a specialized center).”

Authors are comparing and analyzing in the study a diagnostic value of colposcopy performed by experts in the field (gynecologic oncologists). While, in discussion it is described that AI can be used by unexperienced gynecologists and general physicians, as well. Please give a proper discussion on that matter that is in correlation with a study design and presented results as the analysis on high level of colposcopic expertise was compared with the AI application.

> I understand your concern. I have changed the sentences. Considering that the diagnostic value of AI interpretation was comparable to the impressions of colposcopic experts, AI interpretation might be considered as an alternative tool in evaluating high-grade cervical lesions when certified or proficient colposcopists are unavailable.

7) “However, the original intention of the Cerviray AI® system (AIDOT) was to use it for cervical cancer screening in low-income or developing countries where a large number of patients cannot access medical services due to the lack of accessibility or high costs. Therefore, the omission of patients with atypical glandular cells appears to be reasonable.”

It is not very good reason for the omission of AGC, as being country with a very high HDI. Please, explain only in the discussion the limitations of AI in detecting endocervical disease, not only with a glandular origin. Intraepithelial lesions can arise in endocervical canal as well. Is there any other limitation of AI in colposcopy?

>Thank you. I fully agree with your opinion. I have modified the discussions.

Results

1) “Chronic cervicitis/koilocytic atypia”

The histologic diagnosis of LSIL should be established if there was the koilocytic atypia in a morphologic view. Chronic cervicitis and koilocytic atypia have different morphologic features – these two are different histologic diagnoses. There is no chance to do a proper cytologic-histologic correlation using that kind of histologic diagnoses. Please also mention that issue in the Discussion as the limitation of your study.

Please specify also in the Methods, what histology terminology for cervical biopsies interpretation was used in the study. Please also make your introduction correspondent with your results in that matter (LSIL vs CIN terminology).

>I am sorry for the use of wrong terminology. I tended ‘benign findings including chronic cervicitis or koilocytotosis’. This change was applied in the manuscript. Lower anogenital squamous terminology system and Bethsda classification system were used for cytologic and histologic evaluation, respectively. This is specified in method section, and all related terms were changed.

Material and methods vs results

1) “This study included female patients who underwent colposcopy-guided biopsy because of abnormal cervical cytology or a positive HPV status during 2020 at two institutions.”

Authors stated that the study included women with abnormal cervical cytology results or HPV-positive. While, further in the study there is any mention about patients with positive HPV status including in Patient and Characteristics subsection and table 1. If you exclude HPV-positive patients from the study, please mention this in the manuscript. That issue is not clarified enough in the study.

>Sorry for the misses of data about HPV status. I have added HPV status of study population in Table 1.

Conclusions

In the light of abovementioned issues in the review, presented conclusions are not cautious enough and are going too far in AI interpretation as a stand-alone tool in cervical cancer screening. First and foremost, conclusions are not based on presented results. Please, follow the purpose of your study in conclusions.

In summary, caution is needed for AI and precancer and cancer diagnosis. Much more data is needed before we can start using such systems in diagnostics.

>I made a tone-down in conclusions. Thank you for your comments.

References

  1. Ovestad, I.T.; Vennestrom, U.; Andersen, L.; Gudlaugsson, E.; Munk, A.C.; Malpica, A.; Feng, W.; Voorhorst, F.; Janssen, E.A.; Baak, J.P. Comparison of different commercial methods for HPV detection in follow-up cytology after ASCUS/LSIL, prediction of CIN2-3 in follow up biopsies and spontaneous regression of CIN2-3. Gynecol Oncol 2011, 123, 278-283, doi:10.1016/j.ygyno.2011.07.024.

Reviewer 2 Report

Currently there is considerable interest in the use of AI in medical imaging and a number of groups developing AI systems as adjuncts or stand alone assessment.  Such a system would benefit the large number of LMIC introducing cervical screening but without colposcopy capacity to manage screen positive women.

The aim is to develop a system to diagnosis high grade CIN (HGCIN) and as such the study needs to ensure ascertainment of CIN which is a histological diagnosis rather than a comparison with 2 colposcopists.  This is limited in generalisability as it reflects practice of 2 colposcopists in 2 institutions in Republic of Korea.  There are also a limited number of participants and this can only be considered as a pilot study.

The introduction needs to make clear the natural history of CIN and that CIN is a histological diagnosis - not a colposcopic or cytological. Accuracy of colposcopy also depends on the referral population, rate of CIN in these and the number of biopsies taken.  Colposcopy also needs to consider referral results, risk of CIN and risk factors, if the colposcopy is adequate and the type of transformation zone.  Can this system assess adequacy, SCJ and TZ type?  Otherwise the correct clinical decision will not be made.

It will help to understand both the results and the role seen for such technology. also needed is the number of images reviewed, manipulation and different magnifications. 

Of the 234 patients in the study, only 22% had HG cytology but 47% had HGCIN.  This is a high rate in the referral population described and could be better understood if more detail given - where they a selected group or routine referrals?

I am unclear on who the AI was developed and the discussion should be more focused on next steps and clinical contribution

Author Response

I really appreciate for your review of the manuscript. I have applied all the changes you recommended. It was an honor for me to learn your professional point of view regarding detail approach for writing review article. I think this experience would improve my future works.

Thank you.

Best Regards,

Seongmin Kim

The aim is to develop a system to diagnosis high grade CIN (HGCIN) and as such the study needs to ensure ascertainment of CIN which is a histological diagnosis rather than a comparison with 2 colposcopists. 

  • Thank you for your opinion. However, in this study, the diagnostic value of AI was evaluated using histological diagnosis of each patient. Therefore, I think that there is a misunderstanding about study design. All the diagnostic value of AI or colposcopists were calculated regarding the result of colposcopy-guided biopsy.

This is limited in generalisability as it reflects practice of 2 colposcopists in 2 institutions in Republic of Korea.  There are also a limited number of participants and this can only be considered as a pilot study.

  • Thank you for your comment. I agree with you that this study can be considered as a pilot study. I have mentioned about this in the title of the manuscript.

The introduction needs to make clear the natural history of CIN and that CIN is a histological diagnosis - not a colposcopic or cytological.

  • Thank you. I have modified that the introduction section regarding your comment.

 Accuracy of colposcopy also depends on the referral population, rate of CIN in these and the number of biopsies taken.  Colposcopy also needs to consider referral results, risk of CIN and risk factors, if the colposcopy is adequate and the type of transformation zone.  Can this system assess adequacy, SCJ and TZ type?  Otherwise the correct clinical decision will not be made.

  • Thank you for your advice. This AI system can assess adequacy, SCJ and TZ. Additional comments about this are now included in Figure 1 legend.

It will help to understand both the results and the role seen for such technology. also needed is the number of images reviewed, manipulation and different magnifications.

Of the 234 patients in the study, only 22% had HG cytology but 47% had HGCIN.  This is a high rate in the referral population described and could be better understood if more detail given - where they a selected group or routine referrals?

  • The cytology was performed at local clinic or my hospital during medical check-up. The study population only included the patients who had colposcopic examination for certain reason in my hospital. Therefore, the possibility of having high grade lesion could increase due to the nature of referral system in Korea.

I am unclear on who the AI was developed and the discussion should be more focused on next steps and clinical contribution

  • Considering that the diagnostic value of AI interpretation was comparable to the impressions of colposcopic experts, AI interpretation might be considered as an alternative tool in evaluating high-grade cervical lesions when certified or proficient colposcopists are unavailable. The original intention of the Cerviray AI® system (AIDOT) was to use it for cervical cancer screening in low-income or developing countries where a large number of patients cannot access medical services due to the lack of accessibility or high costs.
  • The study population had cytology for their cervical cancer screening. The updated recommendation of primary HPV testing for cervical cancer globally could alter the result. Further study from individuals with regular HPV testing should be performed later.

Round 2

Reviewer 1 Report

It concerns: a review of the manuscript entitled “Role of artificial intelligence interpretation of colposcopic images in cervical cancer screening”. Healthcare-1466151-peer-review (Round 2: Reconsideration after major revision)

In the current version of the manuscript the following issues have been noted after the major revision:

  • serious methodological flaws
  • fundamental errors related to improper colposcopic, and histologic terminology used. Large number of inaccuracies and clear substantive errors in that matter, along with a superficial fulfilling the reviewer’s remarks. Whereas histological diagnoses presented in the study need a deep revision and a new cyto-histologic correlation need to be evaluated.
  • reviewer’s comments have been included very selectively in the revision; some crucial ones have been omitted in the revision.
  • the paper indicates poor QA&QC for gynecological cytopathology (very low cytology sensitivity for histologic high-grade cervical intraepithelial lesions at the HSIL cut-off level!), what have a crucial importance for a comparative analysis of the methods presented in the study and may lead to wrong conclusions.
  • too general methodology for colposcopic AI presented, as it has been proposed as a new diagnostic tool in the study.
  • incoherent actual structure of the manuscript.

Author Response

Responses to reviewer 1 [healthcare-1466151]

Title: Role of artificial intelligence interpretation of colposcopic images in cervical cancer screening:  A pilot study.

Thank you for your advice.

The authors reviewed and revised manuscript regarding your comments.

I am so grateful for you giving me an apportunity to improve the quality of the manuscript.

Thank you.

Best regards,

Seongmin Kim

---------------------------------------------------------------------------

fundamental errors related to improper colposcopic, and histologic terminology used.

  • I have modified the improper terminology. Bethesda classification system and CIN classification system were used for cytologic and histologic evaluation, respectively. The International Federation for Cervical Patholoy and Colposcopy terminology was used for determining colposcopic impression.

Large number of inaccuracies and clear substantive errors in that matter, along with a superficial fulfilling the reviewer’s remarks. Whereas histological diagnoses presented in the study need a deep revision and a new cyto-histologic correlation need to be evaluated.

  • I have reviewed the full data, but there was no error in our data. I could not change the study data this time.

reviewer’s comments have been included very selectively in the revision; some crucial ones have been omitted in the revision.

  • I am sorry for my faults. I had a full review of my data and manuscript again.

the paper indicates poor QA&QC for gynecological cytopathology (very low cytology sensitivity for histologic high-grade cervical intraepithelial lesions at the HSIL cut-off level!), what have a crucial importance for a comparative analysis of the methods presented in the study and may lead to wrong conclusions.

  • This shows a sensitivity at cutoff cytological high-grade lesion including ASC-H or HSIL for detection of histological CIN2 or worse. This could be a reason why the sensitivity is low in this study. In a meta-analysis, the sensitivity of liquid-based cytology and conventional cytology for CIN2 or worse showed 57.1% and 55.2%, respectively [1].

too general methodology for colposcopic AI presented, as it has been proposed as a new diagnostic tool in the study.

  • Additional explanations for terminology were replenished in methods section.

incoherent actual structure of the manuscript.

  • Thank you for your comments. I made an effort to make the meaning and purpose of this study clear.

Reference

  1. Arbyn, M.; Bergeron, C.; Klinkhamer, P.; Martin-Hirsch, P.; Siebers, A.G.; Bulten, J. Liquid compared with conventional cervical cytology: a systematic review and meta-analysis. Obstetrics & Gynecology 2008, 111, 167-177.

Reviewer 2 Report

Thank you for responding to my comments

Author Response

Dear Reviewer 2,

I really appreciate for your generous acceptance of my revised manuscript.

Thank you very much.

Best regards, 

Seongmin Kim